# A Randomized Controlled Trial of Motor Imagery Combined with Virtual Reality Techniques in Patients with Parkinson’s Disease

**DOI:** 10.3390/jpm12030450

**Published:** 2022-03-12

**Authors:** Muhammad Kashif, Ashfaq Ahmad, Muhammad Ali Mohseni Bandpei, Hafiza Aroosa Syed, Ali Raza, Vishal Sana

**Affiliations:** 1University Institute of Physical Therapy, Faculty of Allied Health Sciences, University of Lahore, Lahore 42000, Pakistan; ashfaq.ahmad@uipt.uol.edu.pk (A.A.); mohseni_bandpei@yahoo.com (M.A.M.B.); 2Riphah College of Rehabilitation and Allied Health Sciences, Riphah International University, Faisalabad 38000, Pakistan; hafiza.aroosa.syed@gmail.com (H.A.S.); ali050dpt@gmail.com (A.R.); vishsana24@gmail.com (V.S.); 3Pediatric Neurorehabilitation Research Center, University of Social Welfare and Rehabilitation Sciences, Tehran 1985713871, Iran

**Keywords:** virtual reality, motor imagery, motor function, motor skills, Parkinson’s disease, physical therapy

## Abstract

Background: Parkinson’s disease is the second most common neurological disease, affecting balance, motor function, and activities of daily living. Virtual reality and motor imagery are two emerging approaches for the rehabilitation of patients with Parkinson’s disease. This study aimed to determine the combined effects of virtual reality and motor imagery techniques with routine physical therapy on the motor function components of individuals with Parkinson’s disease. Methods: The study was a prospective, two-arm, parallel-design randomized controlled trial. Forty-four patients with idiopathic Parkinson’s disease were randomly assigned to one of two groups. Virtual reality and motor imagery were given together with physical therapy in the experimental group (N: 20), while physical therapy treatment alone was given in the control group (N: 21). Both groups received allocated treatment for 12 weeks, 3 days a week, on alternate days. Motor function was assessed at baseline, six weeks, twelve weeks, and sixteen weeks after discontinuing treatment with the Unified Parkinson’s Disease Rating Scale part III. SPSS 24 was used to analyze the data. Results: Study results indicate that the experimental group showed significant improvements in the motor function components: tremor at rest at the 6th week (*p* = 0.028), 12th week (*p* = 0.05), and 16th week (*p* = 0.001), rigidity at the 6th week (*p* = 0.03), 12th week (*p* = 0.000), and 16th week (*p* = 0.001), posture at the 12th week (*p* = 0.005) and 16th week (*p* = 0.004), and gait at the 6th week with a *p*-value of (*p* = 0.034). Conclusions: This study demonstrated that virtual reality and motor imagery training in combination with routine physical therapy can significantly improve resting tremors, rigidity, posture, gait, and body bradykinesia in individuals with PD in comparison to patients receiving only routine physical therapy.

## 1. Introduction

Parkinson’s disease (PD) is the second most common chronic neurodegenerative condition, characterized by bradykinesia, stiffness, resting tremors, and postural instability, as well as a variety of motor and non-motor symptoms [1,2]. The Parkinson’s Foundation reports that Parkinson’s disease is a slowly developing disease that affects nearly one million Americans and more than 10 million individuals worldwide [3,4]. Environmental variables, age, and genetic susceptibility are major etiological contributors. The risk of developing PD varies between Asian and non-Asian populations, owing to environmental and genetic factors [5]. The general population prevalence of PD is 0.3 percent, whereas the elderly population is believed to be 1–2 percent [6]. Several studies indicated greater frequency and incidence in males than in females [7].

The primary goal of physical rehabilitation is to assist people with neurological conditions to regain their independence by learning new motor skills and regaining lost motor skills with an increasing emphasis on a client-centered approach [8,9,10]. Non-pharmacological therapy such as exercise programming has been shown to increase strength, physical functioning, quality of life, balance, and gait speed in PD patients [10,11]. PD patients have benefited from the wide range of exercises that are available, including stretching, progressive exercises, aerobic training, relaxation exercises, strength and balance exercises, and treadmill walking, all of which have been studied [11]. However, a few studies evaluating the effects of physical therapy (PT) treatments in different forms have reported a loss of exercise benefit within weeks or months of ceasing the protocol. Furthermore, several barriers to exercise compliance have been identified among PD patients based on concerns regarding longer treatment durations, fear of falling, and financial constraints [12,13].

The scientific explanation to use virtual reality (VR) in rehabilitation originates from the subject of motor learning. VR training is one method that may help generate and maintain user interest [14]. VR training is proving to be a highly effective supplement to traditional therapy for many patient groups. The impacts of introducing VR technology into rehabilitation procedures have been studied before, but they mainly focused on high-end specialized gear and software that is not commonly accessible or inexpensive, making it unsuitable for large-scale clinical or home deployment. To solve these issues, academics have increasingly concentrated on consumer-grade technologies. The Nintendo Wii game system has gotten a lot of academic and clinical attention [15]. A VR system has the potential to provide feedback, an essential aspect of motor learning, and motor relearning can improve brain function via brain plasticity, as well as enhance physical function in elderly people [16] and treat traumatic brain injury [17], vestibular rehabilitation [18], Parkinson’s disease [19], stroke [20], and cerebral palsy [21]. Furthermore, VR has been demonstrated to boost attention span, feelings of achievement, self-esteem, and motivation. Processes of reward motivate game play. During action video game play, significant increases in striatal dopamine levels have been noticed. Dopamine level elevation appears to be associated with enhanced performance, learning, satisfaction, and motivation [22]. VR enhances the significance and relevance of an activity by providing an enhanced environment. As a result of this enhanced engagement with an enriched environment, rehabilitation clients may be better served [23].

Motor imagery (MI) is a cognitive method that requires attention to the sequence of a learned activity, which may be done visually or kinesthetically. Despite abnormalities in the supplementary motor region due to the indirect action of the basal ganglia, individuals with PD have maintained locomotor imagery exhibited during the on-medication state. Although the left hemisphere, namely the posterior parietal cortex, is thought to be essential for movement planning, evidence has shown that motor imagery activates other areas of the brain [24]. During MI tasks and motor execution, the main motor cortex (M1) and secondary motor regions (premotor, supplementary motor, and parietal cortices) are active [25]. MI training has been demonstrated to be useful in the treatment of a variety of disorders with various etiologies, including Parkinson’s disease [26].

Task-oriented training, personalized feedback, goal-tailored exercise regime, regular movement repetition, engaging and exciting gaming situations, personalized treatments, and feedback focused on motor learning are all components of effective rehabilitation programs. VR-based treatment may include all of these components [27,28]. There is evidence that explicit and implicit learning occur concurrently, and that the two may be employed in tandem to increase or complement each other’s impact. The MI is also supposed to support the VR-developed learning process [1]. As a result, it is claimed that MI can promote broad and consolidative learning in PD patients. Both VR- and MI-based therapies employ the use of multiple sensory inputs at the same time [29]. Considering that MI and VR applications are increasingly emerging as potentially useful techniques for rehabilitation in PD, this study was designed to determine whether MI combined with a VR program along with routine PT had an effect on the motor components of individuals with PD, including body bradykinesia, rigidity, resting tremor, postural instability, posture, and gait, in comparison to routine PT alone.

## 2. Materials and Methods

### 2.1. Study Design and Setting

This prospective, randomized single-blinded trial with parallel groups was conducted in 2020–2021 at the Safi Hospital, Faisalabad, Pakistan, in the Department of Physical Therapy.

### 2.2. Study Participants

Patients with PD were recruited from neurology and neurosurgery departments of the Hospital located in Faisalabad city. The patients were subsequently referred to the Department of Physical Therapy Safi Hospital, where patients were evaluated for trial eligibility by a physical therapist. This study included subjects aged 50 to 80 years with idiopathic PD, severity ranging from stage I to stage III on the modified H&Y scale, and intact cognition (mini mental state examination (MMSE) score equal to or greater than 24). Study participants excluded were those with any other neurological or orthopedic condition, visual abnormalities, cardiovascular problems, severe dyskinesia or “on-and-off” phases, prior history of PD surgery, history of virtual game therapy within the past 3 months, or virtual game fear. The neurologist determined the patients’ eligibility based on the above-mentioned criteria.

In order to determine if patients met the inclusion criteria for the experiment, all recruited patients were examined by an independent assessor as part of the baseline assessment. Each participant was evaluated four times during the study: at baseline, at mid-intervention (six and twelve weeks into the trial), and at post intervention one month after the trial had ended. Participants were asked to read the study protocol and sign a permission form prior to the baseline testing session. The participants were encouraged to ask questions and were informed that they might leave the study at any time without consequence. Both interventions and assessments were conducted at the same time of day and two hours after the medication was taken [30,31]. Due to the pharmacodynamics of levodopa (the onset of treatment occurs between 20 and 40 min after taking the medication, and remains for two to four hours), patients with PD were evaluated late in their ON phase [32,33]. All study participants maintained the same medication regimens throughout the study period. We excluded patients with on-off motor fluctuations and dyskinesia above grade 3 on the UPDRS due to the potential for interference with the study results [34].

### 2.3. Sample Size Calculation

The sample size for the study was calculated using the mean Unified Parkinson’s Disease Rating Scale (UPDRS) scores of 25.1 ± 12.8 and 18.5 ± 11.0 for the VR and control groups, respectively, with a confidence interval (α) of 95% and 80% for the study’s power, as retrieved from Yang et al. [35].

### 2.4. Groups and Intervention Procedures

Participants were assigned to either control Group A (only routine PT) or experimental Group B (MI + VR training + routine PT). Group A (control) received only routine PT (such as warm-up, stretching, strengthening, and relaxation exercises; limb coordination exercises; and core, neck, and gait training) along with cycling and walking exercises, while Group B (experimental) received routine PT protocols, as well VR and MI training. There were 22 individuals in each group at the beginning of the study. For 12 weeks, the patients in group A were given 40 min routine PT treatment and 20 min of walking and cycling, with a brief rest time every other day (three days a week). Group B participants had 60 min sessions every other day (three days a week) for 12 weeks, which included 40 min of routine PT, 10–15 min of VR, and 5–10 min of MI. The outcome measurements were collected at the start (week 0), end (weeks 6 and 12), and 1 month following the intervention (week 16).

### 2.5. Randomization and Blinding

The principal investigator (primary author) gave each subject a number, which was then selected randomly from a box, resulted in a random sample using lottery method. During the trial, both groups had a 1:1 participant ratio. A single blinded study with the assessor blinded was conducted owing to the nature of the intervention, which prevented the patients and the principal investigator from being blinded. To ensure an unbiased analysis, the statistician was further blinded by classifying the data into A and B groups. The CONSORT diagram for the study is presented in Figure 1.

### 2.6. Interventions

The interventions used in both the experimental and control groups were based on a previously reported protocol for the rehabilitation of PD with VR and MI, in addition to routine PT treatment [30].

#### 2.6.1. VR Rehabilitation Protocol

Each participant’s VR session lasted 10–15 min. Wii box, Wii controller, and Wii Fit board comprised the VR system. On the Wii Fit board, patients were advised to engage with the VR system and play games. Based on a prior systematic study, three senior physical therapists (movement experts) selected games for three domains: motor functioning, balance, and activities of daily living (ADLs) [33]. The Wii box included many settings ranging from simple to difficult. Two rehearsals were provided to acquaint the patients with the setting and the VR system, as well as to build rapport between the therapist and the participants. The games, the treatment, and the score were described to the patients. In terms of motor functions, tennis, boxing, bowling, and kicking were used, whereas soccer, table tilt, penguin slide, and tilt city were used to enhance dynamic balance, while single-leg extension, and torso twist were used to enhance static balance [36,37,38] (Figure 2).

The Wii Fit board was set up with the patients standing within parallel bars with their shoes removed. There was constant supervision and guidance from the therapist for the patients, as well as timely feedback (if necessary). The VR session began with a series of balancing games. During each training session, there was a dynamic balancing game and a specific static balance game. Exercises were selected based on their difficulty level, and the degree of difficulty was progressively raised in accordance with the patients’ results. They began with the penguin slide and worked their way up through table tilt, tilt city, and then soccer. Each game was first played for 2–3 min. Three to four minutes of table tilt were added as the performers improved. Weight shifts and movement patterns improved as a result of playing this video game. For up to 2 min a day, the patients did single leg extensions. Additional exercises were introduced in subsequent weeks, including soccer, torso twists, and a tilt city. Each session lasted between 1 and 5 min for the individuals to complete these tasks. As the lessons proceeded, they moved on to a variety of motor function sports that ranged from bowling and tennis to kicking and boxing (the latter being the most difficult). The majority of the games might be completed with little or no assistance. Boxing was used in the last 3 weeks of treatment due to an increase in balance and coordination requirements [1].

#### 2.6.2. MI Rehabilitation Protocol

The MI took place in the final 5–10 min of the session and was implemented in three steps. The initial stage was for the participants to view the recorded videos to familiarize with technique. There were two sets of videos: one with normal motions and the other with patients executing the moves. The patients were asked to compare and contrast the two movies. Then, they were told to relax and focus on their peaceful breathing patterns. The participants were instructed on how to sit comfortably and relaxed on a chair having their arms and back supported. Close their eyes and concentrate on slow nasal breath while closing their eyelids. The patients were asked to perform tasks ten times. The individuals were then verbally directed to complete the tasks. During recall, the abnormal movement components were highlighted [1].

#### 2.6.3. Routine PT Treatment

In all, each session lasted 40 min and started with a routine PT session. The patients were first taken through a series of warm-up activities. Patients were asked to inhale and exhale while they sat in a chair with their backs and feet supported. Each exercise was performed five times for a total of five minutes of warm-up. In order to minimize shallow breathing, pushing, and holding one’s breath, the patients were instructed in correct breathing techniques. Supine on the bed, they were instructed to rehearse under the watchful eye of the primary investigator. For 15 min each session, the stretches were held for 10–30 s with four repetitions of each of the following areas: upper chest and neck flexors, shoulder and adductors, elbow and wrist flexions, knee flexions, calves, and lower back, respectively. Each exercise was done 10–15 times throughout each session for a total of 15 min of strength training. Core muscles (abdominals) and hip, knee, back, and elbow extensors were the primary focus of this workout. Slow, prolonged stretches of the shoulder flexors, adductors, and hip and knee flexors were done for 5 min as a cool-down [1,39].

### 2.7. Outcome Measures

#### Assessment of Sub-Components of Motor Function

Motor function of the PD patient was recorded on UPDRS part III by a blind assessor at baseline, 6th and 12th week of therapy, and during follow-up (16th week). UPDRS is a renowned self-report and clinical observation tool frequently used to assess and monitor the progress of patients with PD for motor function using different paradigms. Subscale III of UPDRS was used in this study for rating motor function. The UPDRS rates rigidity, bradykinesia, tremor, and mobility. Excellent internal consistency was found in many studies on UPDRS [40,41]. The UPDRS Section III was designed to offer a comprehensive, efficient, and adaptable way to track the progression of motor symptoms in PD. The scale consists of 14 elements, each graded on a 5-point scale from 0 to 4. The total potential score is 56, with higher values indicating more disability [42]. The number of lost treatments and adverse effects were also recorded as indicators of treatment safety. Moreover, the trial considered these criteria for stopping the intervention: if patients suffer from intolerable or severe adverse events and if the principal investigator determines that the risk outweighs the benefit. After treatment, participants with worsening conditions should be included in the protocol’s efficacy and safety analysis [1].

### 2.8. Statistical Analysis

The primary null hypothesis for the study was that there was no difference in motor function between the VR + MI + routine PT and routine PT alone groups as evaluated by the UPDRS-III.

SPSS version 24 was used for data input and statistical analysis. Quantitative data such as age, gender, age of onset of PD, and PD diagnosis were descriptively analyzed using mean, median, mode, variance, and standard deviation. The normality of the data was determined using the Kolmogorov–Smirnov and Shapiro–Wilk tests. The data from the control and experimental groups did not follow a normal distribution. So, the Mann–Whitney U test was utilized. The changes in mean scores were evaluated to identify which intervention effective. The data were set to *p* < 0.05 significance. Due to loss to follow-up, the experimental group had 20 individuals and the control group had 21.

## 3. Results

A total of 50 subjects were recruited and 44 that fulfilled the eligibility criteria were randomized to routine PT or VR + MI + RP group. Initially, at the 12-week assessment, two individuals from the experimental group and one from the control group were absent due to transportation concerns or caregivers’ unavailability, leaving 20 in the experimental group and 21 in the control group. Baseline characteristics of the population studied showed no significant differences in age with *p*-value (0.936), disease duration with *p*-value (0.887), age at onset of PD with *p*-value (0.912), and age at diagnosis of PD with *p*-value (0.443). The mean score of the H&Y stage in the experimental group was (2.11 ± 0.74), and in the control group it was (2.25 ± 0.67) with *p*-value of (0.720). The mean score of MMSE in the experimental group was (26.41 ± 1.91) and in control group was (25.27 ± 4.38). The mean score of UPDRS-III at baseline in the experimental group was 32.45 ± 3.98, and in the control group it was 31.86 ± 4.62 with *p*-value (0.742) (Table 1).

At baseline, the UPDRS part-III scores showed impaired motor function for all PD patients. When analyzing specific domains of UPDRS part-III, subjects in the experimental group showed a significant improvement in tremor at rest with *p* = 0.028 at 6th week, *p* = 0.005 at 12th week, *p* = 0.001 at 16th week, a significant improvement in rigidity with *p*= 0.003 at 6th week, *p* < 0.001 at 12th week, *p* = 0.001 at 16th week, a significant improvement in posture with *p* = 0.005 at 12th week, *p* = 0.001 at 16th week, a significant improvement in postural stability with *p* = 0.220 at 6th week, *p* = 0.013 at 12th week, *p* = 0.042 at 16th week, and a significant improvement body bradykinesia with *p* = 0.088 at 6th week, *p* = 0.026 at 12th week, *p* = 0.035 at 16th week (Figure 3).

No differences in outcomes including the speech, fascial expression, action or postural tremor, finger taps, hand movements, or leg agility analyzed were observed between the two groups. However, a significant improvement in gait with *p* = 0.032 at 6th week, *p* = 0.001 at 12th week, *p* = 0.03 at 16th week, a significant improvement in rapid alternating movements with *p* = 0.047 at 12th week, *p* = 0.027 at 16th week, and a significant improvement in arising from a chair with *p* = 0.012 at 12th week, *p* = 0.016 at 16th week, were revealed in the experimental group (Table 2).

In the experimental group, the mean and SD for speech were (2.00 ± 0.000) at baseline, (1.86 ± 0.351) at 6 weeks, (1.64 ± 0.492) at 12 weeks, and (1.82 ± 0.395) at follow up. In the control group, mean score was (1.95 ± 0.13) at baseline, (1.95 ± 0.213), after 6 weeks (1.82 ± 0.395), after 12 weeks (1.59 ± 0.666), and after 16 weeks (1.59 ± 0.796), respectively. Resting tremor mean and SD in the experimental group after 16 weeks was (3.31 ± 1.21) and in the control group it was (4.86 ± 1.28). Action or postural tremor mean and SD in the experimental group at baseline was (2.00 ± 0.53), 12 weeks was (0.77 ± 0.611), and 16 weeks was (1.18 ± 0.664), and in the control group at baseline it was (1.77 ± 0.102), 12 weeks was (1.18 ± 0.795), and 16 weeks was (1.40 ± 0.734). In the experimental group, the mean and SD of rigidity at baseline and at follow up were (4.86 ± 0.710) and (2.27 ± 1.55), respectively, whereas in the control group it was (4.68 ± 0.893), and (4.09 ± 1.68) with a *p*-value < 0.001. In the experimental group, the mean and SD for gait were (1.86 ± 0.468) at baseline, (1.23 ± 0.429) at 6 weeks, (0.86 ± 0.560) at 12 weeks, and (1.00 ± 0.690) at 16 weeks, whereas in the control group, they were (1.73 ± 0.456) at baseline, (1.55 ± 0.510) at 6 weeks, (1.50 ± 0.598) at 12 weeks, and (1.45 ± 0.671) at 16 weeks. A comparison of the mean and SD for postural stability in the experimental and control groups at baseline was (1.50 ± 0.512), after 6 weeks was (1.14 ± 0.468), after 12 weeks was (0.77 ± 0.685), and after 16 weeks was (0.73 ± 0.631). The mean and SD for body bradykinesia in the experimental group after 16 weeks was (0.95 ± 0.653), and in the control group after 16 weeks it was (1.36 ± 0.727) (Table 3).

## 4. Discussion

To the best of our knowledge, this is the first trial to show the effects of Nintendo Wii virtual games and MI along with routine PT on the components of motor function such as tremors, posture, gait, body bradykinesia, and postural instability in PD patients. The current study demonstrates that PD patients with VR and MI exhibit significant improvements in several motor components, such as tremors, posture, body bradykinesia, and postural stability. In the same vein, a recent case study found that VR combined with MI and PT improved balance, motor function, and ALDs in two patients with PD [43].

The combination of MI and PT reduces bradykinesia, as reported by the current study. Research by Lokhandwala indicated that combining motor imagery with actual practice may be more helpful in treating Parkinson’s disease, particularly bradykinesia. However, there was no significant improvement in UPDRS across groups. The study revealed that combining mental and physical activity is superior to physical practice alone. Thus, combined treatment excelled physical exercise alone in lowering bradykinesia in Parkinson’s patients [44] Robles-García conducted an RCT on the effects of imitation treatment in Parkinson’s. In total, 16 patients were divided into experimental and control groups. Four weeks of finger tapping with the dominant hand were used to differentiate groups. Self-paced movement characteristics and cortico-spinal excitability were assessed before, during, and after training. Both the trained and untrained hands’ movement amplitude increased considerably following treatment. The study found that utilizing VR to imitate movement improved motor practice in Parkinson’s patients, which may help reduce bradykinesia [45]. Motor imagery, according to Abbruzzese et al., alters movement speed and execution. The influence of MI speed on real movement length was studied throughout a 3-week training period. The participants were instructed to mentally execute a sequence of physical actions at varying rates. The impact of MI on real speed execution supports the ideo-motor hypothesis. They say the benefits of mental preparation on physical performance stem from the strong relationship between motor rehearsal and real performance [46].

Our study revealed a significant improvement in resting tremor in patients in the experimental group. Cornacchioli et al. (2020) used the oculus rift virtual reality system to evaluate its effectiveness on resting hand tremors. Participants used the oculus rift S head and touch controllers with specially designed algorithms to eliminate hand tremors. The virtual environment minimized hand tremors in all patients during the task. After the experiment ended, they noted a significant decrease in hand tremors in one of their patients with early PD [47]. Helmich et al. studied the effects of motor imagery on Parkinson’s tremor patients. The disease severity ratings of 20 Parkinson’s patients with tremors and 20 Parkinson’s patients without tremors were compared. The tremor group exhibited higher resting and postural tremor than the non-tremor group, but action tremor intensity did not vary. Non-tremor PD patients reported significantly greater axial, gait, speech, and hypomimia symptoms. The study concluded that PD patients with tremor performed better behaviorally than those without. Decreased imagery-related activity in the somatosensory region was seen in both tremor-positive and -negative PD patients. Tremors in PD patients improved significantly [48].

Virtual reality games improved gait, postural balance, quality of life, and vestibular functioning in Parkinson’s patients, according to Severiano et al. (2018) [49].

According to the findings of our study, the gait of participants in the experimental group has greatly improved. The findings of another research indicated that MI and VR are effective therapeutic techniques for individuals suffering from balance and gait disorders, such as Parkinson’s disease, since these patients demonstrate better balance and gait after receiving the therapy [31]. In a comprehensive study, Triegaardt and colleagues investigated that VR rehabilitation surpassed active treatment in improving gait speed, balance, coordination, quality of life, and functional capacity. Moreover, as compared to a passive intervention. VR improved balance and gait speed. A thorough review indicated that VR improves balance, coordination, motor function, and quality of life. Using virtual reality to improve outcomes in Parkinson’s patients has a substantial impact [50]. One study found that VR improved gait measures including step and stride length. In PD, slowing gait speed, increasing gait diversity, and increasing double-stance time are common [51]. However, in PD, the automatic gait control system diminishes, necessitating attention approaches [52]. VR may increase stride amplitude by giving more precise motor input. According to a recent study, VR treadmill training enhanced gait speed and step length [31]. Other PD symptoms such as postural instability respond well to dopaminergic therapy. So, PT with or without VR may assist postural instability [53,54]. De Melo and colleagues studied VR and gait training’s impact on walking distance and overall fitness in Parkinson’s patients. The VR and treadmill groups outperformed the control group in terms of total distance and gait speed, as well as a pre-6-min walk test of heart rate (HR). The study found that integrating VR and gait training increased walking distance and gait adjustments in Parkinson’s patients [55]. Abraham, Duncan, and Earhart (2021) studied motor imagery in Parkinson’s neurorehabilitation, focusing on gait, balance, and pain impairments. Several cognitive pathways, such as attentional concentration and body schema, are used to ameliorate motor and non-motor symptoms. They found that motor imaging might help Parkinson’s sufferers’ motor functions [56].

Although no prior research has looked at the combined benefits of VR and MI treatment in individuals with PD except as a recently published case report [43], both approaches have been utilized separately to treat other neurological disorders. Patients with various neurological problems have improved in many trials, although the results in these people seem to be better across diverse training regimens [57]. This impact may be explained in the present research by the increased demands imposed on implicit and explicit memory systems. When opposed to other technical improvements, another advantage of this combination technique is the performance of the original movement pattern. The irregular movement timings and coordination abnormalities in these individuals cause mobility difficulties and limits. In contrast to other neurological disorders, muscle weakness and limited range of motion play a secondary role in movement limits. Coordination examinations are seldom done on a regular basis in PD patients, and coordination deficiencies are not easily corrected by manual procedures. Inadequate mobility and aberrant movement patterns are the outcomes of these impairments [58]. In a combined approach, VR and MI largely aid in the normalization of movement start and completion patterns. These cutting-edge strategies also help patients adjust unproductive motions and actively avoid them when the situation calls for them. Persons with PD have a restricted capacity to learn new activities and employ new movement patterns. Maybe the MI technique’s extra motor learning impact on the care plan is to blame. The diverse components of motor learning efforts come into play when VR and MI are combined, improving and adding to the effects of these two distinctive therapies.

## 5. Limitations

There are a few limitations to this study that need to be taken into account by other studies in the future. First of all, this study examined a small sample of PD patients who had mild-to-moderate symptoms. With VR and computer-controlled cognitive rehabilitation, patients with cognitive issues can improve their visual attention and spatial cognition. Participants with cognitive impairment were excluded from this study since they had to follow both the game instructions as well as the verbal commands. As a consequence, a study that takes a holistic approach, concentrating not only on motor but also cognitive impairment, may provide superior findings. VR along with MI might be more effective than conventional therapy; however, further studies with standardized protocols are necessary. The current study is one of the first to incorporate both VR and MI techniques in addition to routine PT. On the basis of prior research, it is anticipated that the combination therapy will be superior to the control therapy. Thus, the intriguing question is whether combination therapy is more beneficial than VR or MI alone. Unfortunately, the current study design precludes answering this question, which is a significant shortcoming of the study. Thus, additional research should be undertaken to compare VR with MI in people with PD.

## 6. Conclusions

The findings of this study revealed that patients receiving VR+MI training in addition to routine PT showed significant improvements in resting tremors, rigidity, gait, posture, body bradykinesia, arising from a chair, and rapid alternating movements, compared to patients assigned to a control group that received only physical therapy. Furthermore, the gains were sustained at follow-up in the experimental group. Thus, VR+MI training + routine PT exercise might be the most effective in treating older adults with mild-to-moderate PD stages.

## Figures and Tables

**Figure 1 jpm-12-00450-f001:**
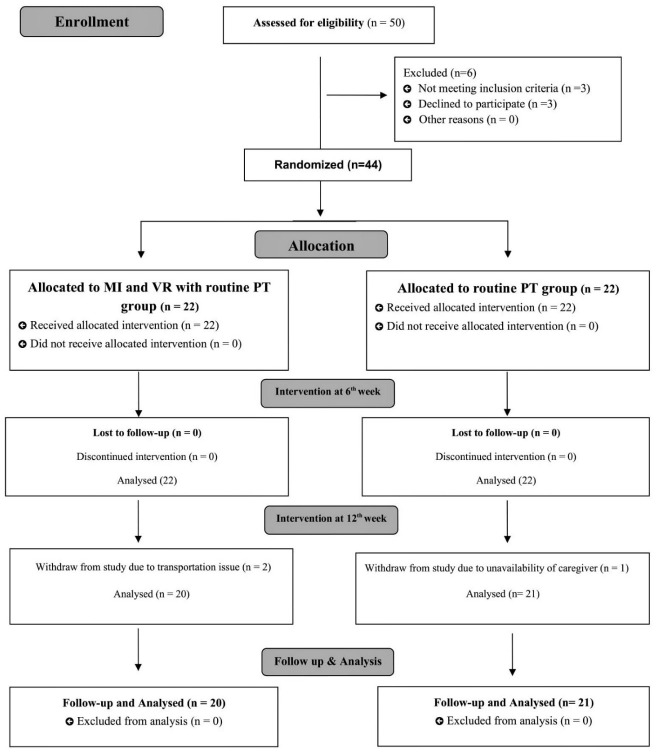
CONSORT study flow diagram.

**Figure 2 jpm-12-00450-f002:**
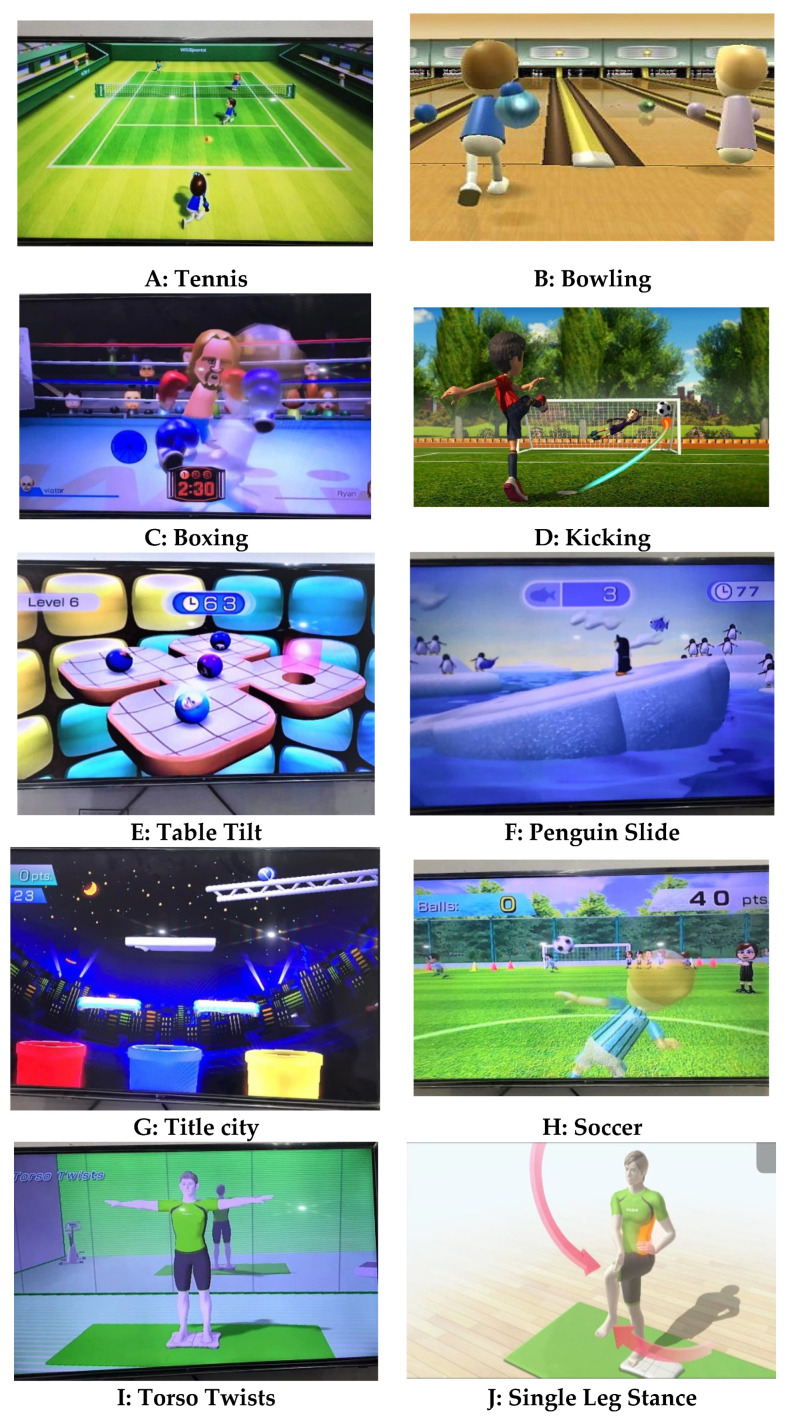
Games used in VR training.

**Figure 3 jpm-12-00450-f003:**
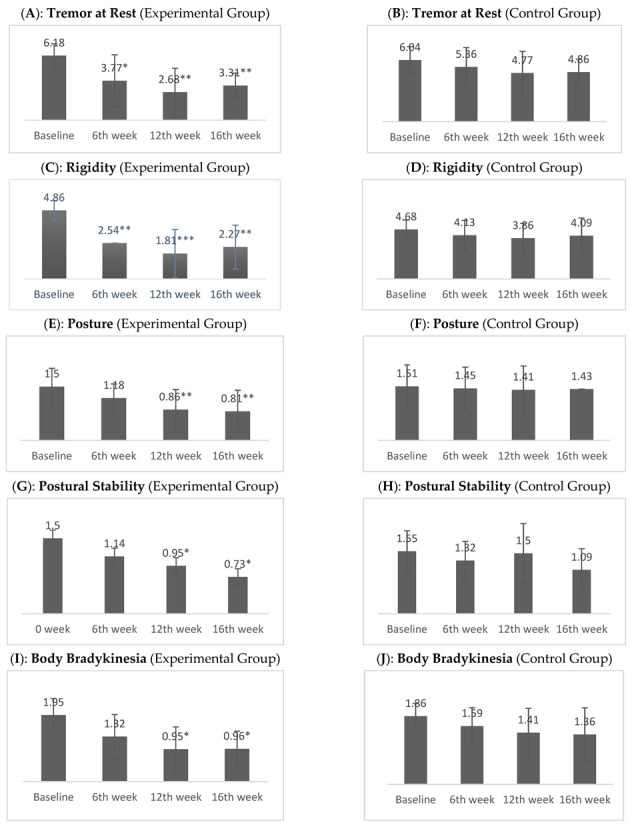
Difference between mean scores of experimental and control groups on UPDRS part III. Note. * *p* < 0.05, ** *p* < 0.01, *** *p* < 0.001.

**Table 1 jpm-12-00450-t001:** Demographics and clinical characteristics of the study subjects

	Randomized (*n* = 44)	*p*-Value
Variables	Experimental/Control Group	
	(*n* = 22)/(*n* = 22)	
Age (years)	63.86 ± 4.57/62.32 ± 4.61	0.936
Gender		
Female	9 (41%)/10 (45.45%)	
Male	13 (59%)/12 (54.55%)	
Height (cm)	160.36 ± 3.70/164.36 ± 2.68	0.397
Weight (kg)	59.59 ± 4.90/60.73 ± 5.43	0.71
Disease duration (years)	6.23 ± 1.85/6.55 ± 1.68	0.887
Age at onset of PD	56.00 ± 4.06/55.50 ± 4.53	0.912
Age at diagnosis of PD	59.55 ± 3.91/60.05 ± 4.13	0.443
H&Y Stage	2.11 ± 0.74/2.25 ± 0.67	0.72
MMSE	26.41 ± 1.91/25.27 ± 4.38	0.029
UPDRS-III baseline	32.45 ± 3.98/31.86 ± 4.62	0.742

PD; Parkinson’s disease, MMSE; Mini mental state examination, H&Y: Hoehn and Yahr Stage.

**Table 2 jpm-12-00450-t002:** Comparison of experimental and control group regarding the mean scores of UPDRS part III.

Motor Function	Groups	Baseline	Assessment at 6th Week	Assessment at 12th Week	Follow Up at 16th Week
		Mean ± SD	Mean ± SD	Mean ± SD	Mean ± SD
Speech	Exp.	2.00 ± 0.000	1.86 ± 0.351	1.27 ± 0.631	1.76 ± 0.429
	Control	1.95 ± 0.213	1.82 ± 0.395	1.36 ± 0.658	1.51 ± 0.740
	Z	−1	−0.407	−0.523	−0.634
	*p* Value	0.317	0.684	0.601	0.526
Facial Expression	Exp.	1.95 ± 0.213	1.91 ± 0.294	1.64 ± 0.492	1.68 ± 0.477
	Control	1.95 ± 0.213	1.82 ± 0.395	1.64 ± 0.492	1.59 ± 0.590
	Z	0	−0.869	0	−0.412
	*p* Value	1	0.385	1	0.68
Action or Postural Tremor	Exp.	2.00 ± 0.53	1.27 ± 0.882	0.77 ± 0.611	1.18 ± 0.664
	Control	1.77 ± 1.02	1.45 ± 0.962	1.18 ± 0.795	1.40 ± 0.734
	Z	−1.392	−0.482	−1.779	−1.02
	*p* Value	0.164	0.63	0.075	0.308
Finger Taps	Exp.	2.23 ± 0.528	1.81 ± 0.501	1.18 ± 0.795	1.45 ± 0.670
	Control	2.18 ± 0.501	1.95 ± 0.722	1.86 ± 0.710	1.72 ± 0.882
	Z	−0.309	−0.956	−1.472	−1.391
	*p* Value	0.757	0.339	0.141	0.164
Hand Movements	Exp.	1.54 ± 0.509	1.13 ± 0.833	0.72 ± 0.882	0.86 ± 0.940
	Control	1.45 ± 0.738	1.18 ± 0.852	1.09 ± 0.811	1.00 ± 0.872
	Z	−0.395	−0.075	−1.47	−0.524
	*p* Value	0.693	0.94	0.142	0.6
Rapid Alternating Movements	Exp.	1.78 ± 0.428	1.40 ± 0.59	1.00 ± 0.690	1.09 ± 0.610
	Control	1.86 ± 0.467	1.59 ± 0.503	1.40 ± 0.59033	1.50 ± 0.51
	Z	−0.629	−1.007	−1.989	−2.209
	*p* Value	0.53	0.314	0.047	0.027
Leg Agility	Exp.	1.91 ± 0.683	1.27 ± 0.882	0.863 ± 0.83	1.00 ± 0.872
	Control	1.90 ± 0.81	1.63 ± 1.04	1.36 ± 1.09	1.32 ± 1.13
	Z	−0.129	−1.229	−1.522	−0.909
	*p* Value	0.898	0.219	0.128	0.363
Arising from a Chair	Exp.	1.18 ± 0.395	0.91 ± 0.526	0.50 ± 0.598	0.50 ± 0.598
	Control	1.23 ± 0.528	1.09 ± 0.684	1.05 ± 0.722	1.00 ± 0.690
	Z	−0.405	−0.989	−2.515	−2.407
	*p* Value	0.685	0.323	0.012	0.016
Gait	Exp.	1.86 ± 0.468	1.23 ± 0.429	0.86 ± 0.560	1.00 ± 0.690
	Control	1.73 ± 0.456	1.55 ± 0.510	1.50 ± 0.598	1.45 ± 0.671
	Z	−0.935	−2.143	−3.273	−2.171
	*p* Value	0.35	0.032	0.001	0.03

Exp.: Experimental, SD: standard deviation.

**Table 3 jpm-12-00450-t003:** Within group comparison of mean scores of UPDRS-III.

Motor Function	Groups	Baseline	Assessment at 6th Week	Assessment at 12th Week	Follow Up at 16th Week	Friedman Test	
		Mean ± SD	Mean ± SD	Mean ± SD	Mean ± SD	X^2^	*p*
Speech	Exp.	2.00 ± 0.000	1.86 ± 0.351	1.27 ± 0.631	1.76 ± 0.429	28.933	<0.001
	Control	1.95 ± 0.213	1.82 ± 0.395	1.36 ± 0.658	1.51 ± 0.740	20.068	<0.001
Facial Expression	Exp.	1.95 ± 0.213	1.91 ± 0.294	1.64 ± 0.492	1.68 ± 0.477	17.077	0.001
	Control	1.95 ± 0.213	1.82 ± 0.395	1.64 ± 0.492	1.59 ± 0.590	12.636	0.005
Tremor at Rest	Exp.	6.18 ± 1.14	3.77 ± 2.50	1.68 ± 1.08	3.31 ± 1.21	46.523	<0.001
	Control	6.04 ± 1.32	5.36 ± 1.91	4.90 ± 1.97	4.86 ± 1.28	16.804	0.001
Action or Postural Tremor	Exp.	2.00 ± 0.53	1.27 ± 0.882	0.77 ± 0.611	1.20 ± 0.664	35.318	<0.001
	Control	1.77 ± 1.02	1.45 ± 0.962	1.18 ± 0.795	1.40 ± 0.734	7.487	0.058
Rigidity	Exp.	4.86 ± 0.710	2.54 ± 1.68	1.81± 1.70	2.27 ± 1.55	43.219	<0.001
	Control	4.68 ± 0.893	4.13 ± 1.42	3.86 ± 1.39	4.09 ± 1.68	15.518	0.001
Finger Taps	Exp.	2.23 ± 0.528	1.81 ± 0.501	1.18 ± 0.795	1.45 ± 0.670	33.444	<0.001
	Control	2.18 ± 0.501	1.95 ± 0.722	1.86 ± 0.710	1.72 ± 0.882	8.202	0.042
Hand Movements	Exp.	1.54 ± 0.509	1.13 ± 0.833	0.72 ± 0.882	0.86 ± 0.940	30.429	<0.001
	Control	1.45 ± 0.738	1.18 ± 0.852	1.09 ± 0.811	1.00 ± 0.872	11.949	0.008
Rapid Alternating Movements	Exp.	1.78 ± 0.428	1.40 ± 0.59	1.00 ± 0.690	1.09 ± 0.610	31.33	<0.001
	Control	1.86 ± 0.467	1.59 ± 0.503	1.40 ± 0.590	1.50 ± 0.51	20.085	<0.001
Leg agility	Exp.	1.91 ± 0.683	1.27 ± 0.882	0.863 ± 0.83	1.00 ± 0.872	32.385	<0.001
	Control	1.90 ± 0.81	1.63 ± 1.04	1.36 ± 1.09	1.32 ± 1.13	20.542	<0.001
Arising from a Chair	Exp.	1.18 ± 0.395	0.91 ± 0.526	0.50 ± 0.598	0.50 ± 0.598	27.117	<0.001
	Control	1.23 ± 0.528	1.09 ± 0.684	1.05 ± 0.722	1.00 ± 0.690	4.017	0.26
Posture	Exp.	1.50 ± 0.512	1.18 ± 0.395	0.86 ± 0.560	0.82 ± 0.588	32.941	<0.001
	Control	1.50 ± 0.598	1.45 ± 0.596	1.41 ± 0.666	1.45 ± 0.596	2.4	0.494
Gait	Exp.	1.86 ± 0.468	1.23 ± 0.429	0.86 ± 0.560	1.00 ± 0.690	36.609	<0.001
	Control	1.73 ± 0.456	1.55 ± 0.510	1.50 ± 0.598	1.45 ± 0.671	11.308	0.01
Postural Stability	Exp.	1.50 ± 0.512	1.14 ± 0.468	0.95 ± 0.653	0.73 ± 0.631	22.983	<0.001
	Control	1.55 ± 0.510	1.32 ± 0.477	1.50 ± 0.740	1.09 ± 0.526	10.220	0.017
Body Bradykinesia	Exp.	1.95 ± 0.486	1.32 ± 0.646	0.95 ± 0.653	91 ± 0.684	44.146	<0.001
	Control	1.86 ± 0.351	1.59 ± 0.503	1.41 ± 0.666	1.36 ± 0.727	21.682	<0.001

## Data Availability

Data generated or analyzed during the research are described in this article. For further information, please contact the corresponding author.

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
