# Peer review of "A Randomized Controlled Trial of Motor Imagery Combined with Virtual Reality Techniques in Patients with Parkinson’s Disease"

_jpm, 2022, doi:10.3390/jpm12030450_

Round 1
Reviewer 1 Report
Good job. The only thing to correct: there is no data relating to drug therapy taken by patients. It should also be specified whether the LEDD or drug plan was changed during the 12 weeks of treatment. IF this were the case, there would be an obvious bias
Author Response
On behalf of our co-authors, We thanks you for your very helpful comments and support for our manuscript.
Comments and Suggestions for Authors
Good job. The only thing to correct: there is no data relating to drug therapy taken by patients. It should also be specified whether the LEDD or drug plan was changed during the 12 weeks of treatment. IF this were the case, there would be an obvious bias
Author’s Response: The suggested information are incorporated in the manuscript
Reviewer 2 Report
Peer review jpm-1614452
Parkinson’s disease (PD) is one of the most common neurodegenerative disorders and severely affects patients’ quality of life as motor function is severely impaired. There is ample evidence that virtual reality (VR) and motor imagery (MI) are beneficial treatment paradigms to improve motor function of PD patients. In the present study, Kashif et al. determine the effects of a combinatorial treatment approach combining VR and MI (experimental group in comparison to control treatment consisting of physical training only). The experimental group showed significant improvements especially in tremor at rest, posture and gait demonstrating that VR in combination with MI together with physical training is superior to physical training only.
The study could be of interest, since it contributes to our understanding of how treatment could be effectively designed and also tailored towards individual needs to improve motor function in PD patients. However, there are several points of concern in the study which might be worth to consider to improve the manuscript:
Broad comments
- As the authors mention in the introduction and the discussion, results from several previous studies indicate that VR and MI, respectively, are beneficial approaches in PD treatment. The present study is one of the first studies combining both strategies along with physical exercise. Based on previous findings, it is expected that the combination therapy is superior to the control therapy. Therefore, the interesting point would be if combination therapy has more benefits than VR or MI alone. Unfortunately, the current study design does not allow to answer this question, which is a major pitfall of the study. This should be appropriately addressed in the “limitations” section.
- Some results of the study are quite impressive considering that the VR+MI treatment strategy significantly improved specific motor function parameters. However, the presentation of the data could be majorly improved to help the reader navigate the findings and evaluate the meaning of the results. This could be done by reorganizing the results section and highlighting significant findings, for example by writing p-values <0.05 in bold in the Tables and adding p-values to Figure 3 (please also see specific comments). Furthermore, it would be beneficial to shorten the sentences and incorporate the numbers in a clearer way.
- The manuscript would benefit from major language editing. Some sentences are hard to follow due to grammatical errors or incorrect wording. Please find specific examples and suggestions in the specific comments.
Specific comments
- The text includes several abbreviations which are not explained before. Please write the complete term fist before using the abbreviation, for example:
- Line 27/60: PT
- Line 173: ADL
- Line 325: RCT
- Line 375: HR
- Unclear wording/definition:
- Line 123: what does “transfer independently” mean?
- Line 159: What does “impartial” mean in this context?
- Line 200: Please describe the “recorded footage” in more detail.
- Line 234: What does “lost treatment” mean
- Line 399-400: sentence is unclear in the context
- Methods:
3.1: The numbers stated in the sample size calculation (line 139) are not clear.
3.2: line 143, 144, 147: Use similar terms to describe treatment paradigms (RPT vs. RP vs. Routine PT)
3.3: line 206: What specifically where the tasks the participants were asked to do?
3.4: The section “Outcome measures” (2.7) has only one subsection (2.7.1). Please remove the numbering of this subsection.
3.5: Line 237: the listing of reasons to stop the treatment starts with 1., but there is no 2.. Please revise this paragraph.
4) Figure 1:
- the box below “enrollment” on the left side of the arrow can be removed since it is the same as on the right side.
- Correct title in the box below “allocation” on the right: remove “with”
- Correct title in the boxes below “Intervention at 12th week”: remove “from” after “study”
- Choose similar title for left and right boxes below “Follow up & Analysis”
- Results:
5.1: line 259-262: Please incorporate p-values correctly into the sentence structure
5.2: line 261: p-value for age at onset of PD differs from the respective number in Table 1
5.3: the MMSE score differs significantly between control and experimental group (p=0.029). This should also be mentioned in the text, not only in Table 1.
5.4 UPDRS part III scores at baseline are missing in Table 1. Please add this information since it is the critical parameter analyzed in this study.
5.5 The authors only provide a detailed description of the effect of VR+MI on different parameters of the UPDRS score, however, a similar description for the outcome in the control group is missing. Please add some information about the results in the control group.
5.6: Line 294-297: the numbers and commas in this sentence are very confusing and it is hard for the reader to follow. It would be beneficial to reorganize the sentence.
- Figure 2: J) shows “single leg stance”, however, in the text only single leg extension is mentioned. Is it the same? If so, please use the same term.
- Figure 3:
- Figure Legend: replace “means” by “mean”
- It would be beneficial to add significant p-values in the graphs
- Table 2: Spelling error: “Leg Aglity”
- Conclusion: in line 417 it says “control group not receiving physical therapy”. I think this is not correct as the control group did receive physical therapy but no VR and MR training
- Grammar:
- Line 55: replace “regaining” by “to regain”
- Line 79: punctuation
- Line 82: replace “associated to” by “associated with”
- Line 110: add “on” after “effect”
- Line 111: remove “as” after “including”
- Line 112: typo (“postural”)
- Line 129: revise sentence
- Line 251-252: revise sentence
- Line 288-290: revise sentence
- Line 371: add “as” after “such”
- Sentence structure:
- Line 47: add “individuals” after “10 million”
- Line 60: Incorporate “however” into the sentence
- Line 64: sentence is incomplete
- Line 73 – 75: revise sentence
- Line 179: revise sentence
- Line 205-206: sentence is incomplete
- Line 273-275: Please complete the sentence
- Line 303-309: Please revise sentences
- Wording:
- Line 50-51: revise sentence
- Line 56: replace “exercise programming” by “exercise program”
- Line 75: replace “engine” by “motor”
- Line 81: add “levels” after “dopamine”
- Line 104-106: Please revise sentence
- Line 109: replace “after” by “in”
- Line 134: replace “before” with “prior”
- Line 158: do the authors mean “unavailability” instead of “availability”?
- Line 339: replace “trement” by “tremor”
- Line 380-81: revise sentence
- Citations: Throughout the manuscript citations are often not implemented correctly, for example:
- citation numbers appear at the beginning of the following sentence after full stop (line 47, 83, 368, 372)
- year of publication is missing after the author names (line 332, 345)
- Reference list: Reference 26 is invalid
Author Response
Dear Reviewer:
We thanks for your very helpful comments and support for our manuscript. The suggested changes are incorporated in the manuscript
Suggestions for Authors
Peer review jpm-1614452
Parkinson’s disease (PD) is one of the most common neurodegenerative disorders and severely affects patients’ quality of life as motor function is severely impaired. There is ample evidence that virtual reality (VR) and motor imagery (MI) are beneficial treatment paradigms to improve motor function of PD patients. In the present study, Kashif et al. determine the effects of a combinatorial treatment approach combining VR and MI (experimental group in comparison to control treatment consisting of physical training only). The experimental group showed significant improvements especially in tremor at rest, posture and gait demonstrating that VR in combination with MI together with physical training is superior to physical training only.
The study could be of interest, since it contributes to our understanding of how treatment could be effectively designed and also tailored towards individual needs to improve motor function in PD patients. However, there are several points of concern in the study which might be worth to consider to improve the manuscript:
Broad comments
- As the authors mention in the introduction and the discussion, results from several previous studies indicate that VR and MI, respectively, are beneficial approaches in PD treatment. The present study is one of the first studies combining both strategies along with physical exercise. Based on previous findings, it is expected that the combination therapy is superior to the control therapy. Therefore, the interesting point would be if combination therapy has more benefits than VR or MI alone. Unfortunately, the current study design does not allow to answer this question, which is a major pitfall of the study. This should be appropriately addressed in the “limitations” section.
- Some results of the study are quite impressive considering that the VR+MI treatment strategy significantly improved specific motor function parameters. However, the presentation of the data could be majorly improved to help the reader navigate the findings and evaluate the meaning of the results. This could be done by reorganizing the results section and highlighting significant findings, for example by writing p-values <0.05 in bold in the Tables and adding p-values to Figure 3 (please also see specific comments). Furthermore, it would be beneficial to shorten the sentences and incorporate the numbers in a clearer way.
- The manuscript would benefit from major language editing. Some sentences are hard to follow due to grammatical errors or incorrect wording. Please find specific examples and suggestions in the specific comments.
Authors’ response: We are thankful to review for valuable comments. The suggested are incorporated in the manuscript.
Specific comments
- The text includes several abbreviations which are not explained before. Please write the complete term fist before using the abbreviation, for example:
- Line 27/60: PT
Authors’ response: The abbreviation is added in the manuscript
- Line 173: ADL
Authors’ response: The abbreviation is added in the manuscript
- Line 325: RCT
Authors’ response: The abbreviation is added in the manuscript
- Line 375: HR
Authors’ response: The abbreviation is added in the manuscript
- Unclear wording/definition:
- Line 123: what does “transfer independently” mean?
Authors’ response: The correction is made in the manuscript for clarity
- Line 159: What does “impartial” mean in this context?
Authors’ response: The correction is made in the manuscript for clarity
- Line 200: Please describe the “recorded footage” in more detail.
Authors’ response: The suggested changes are incorporated in the manuscript.
- Line 234: What does “lost treatment” mean
Authors’ response: The correction is made in the manuscript for clarity
- Line 399-400: sentence is unclear in the context
Authors’ response: The sentence is rewritten for clarity
- Methods:
3.1: The numbers stated in the sample size calculation (line 139) are not clear.
Authors’ response: The sentence is rewritten for clarity
3.2: line 143, 144, 147: Use similar terms to describe treatment paradigms (RPT vs. RP vs. Routine PT)
Authors’ response: The same terms are added throughout the manuscript as suggested by reviewer.
3.3: line 206: What specifically where the tasks the participants were asked to do?
Authors’ response: The required information are incorporated in the manuscript
3.4: The section “Outcome measures” (2.7) has only one subsection (2.7.1). Please remove the numbering of this subsection.
Authors’ response: The suggested changes are incorporated in the manuscript
3.5: Line 237: the listing of reasons to stop the treatment starts with 1., but there is no 2.. Please revise this paragraph.
Authors’ response: The suggested corrections are incorporated in the manuscript
4) Figure 1:
- the box below “enrollment” on the left side of the arrow can be removed since it is the same as on the right side.
Authors’ response: The suggestion correction incorporated in the manuscript
- Correct title in the box below “allocation” on the right: remove “with”
Authors’ response: The suggested corrections are incorporated in the manuscript
- Correct title in the boxes below “Intervention at 12thweek”: remove “from” after “study”
Authors’ response: The suggested corrections are incorporated in the manuscript
- Choose similar title for left and right boxes below “Follow up & Analysis”
Authors’ response: The suggested corrections are incorporated in the manuscript
- Results:
Round 2
Reviewer 2 Report
I would like to thank the authors for addressing most of the reviewer’s comments. The manuscript has majorly improved in terms of readability and clarity. However, there are still some points of concern that are outlined below:
1) Figure 3:
-
- The p-values have been added to the graphs as recommended. However, the graph now is quite unclear. In my opinion, a better way to indicate significant changes would be to use stars above the respective graphs (*<p0.05, **<0.01…)
- In Figure 3E the sizes of the bars in the bar graph do not match the numbers indicated above the bars as in the experimental group the third and fourth bar are higher than the second, third and fourth bar in the control group although they represent a lower value (0.86/0.81 exp. group vs. 1.45/1.41/1.43 ctrl group)
- The graphs in Figure 3A and B are not correctly aligned
2) Conclusion:
To be more concise, I would recommend defining “experimental group” Line 517-18) in more detail, e.g. “patients receiving VR+MI training in addition to routine PT”.
3) Language:
3.1 Line 251: replace “familiar” by “familiarize”
3.2 Line 328-335: remove “the” before “significant improvement”
3.3 Line 478: remove “as”
4) Citation 27 is invalid.
Author Response
Response to Reviewer Comments
Dear Reviewer:
We thanks you for your very valuable feedback, comments and support for our manuscript. The suggested changes are incorporated in the manuscript
I would like to thank the authors for addressing most of the reviewer’s comments. The manuscript has majorly improved in terms of readability and clarity. However, there are still some points of concern that are outlined below:
1) Figure 3:
- The p-values have been added to the graphs as recommended. However, the graph now is quite unclear. In my opinion, a better way to indicate significant changes would be to use stars above the
- respective graphs (*<p0.05, **<0.01).
- In Figure 3E the sizes of the bars in the bar graph do not match the numbers indicated above the bars as in the experimental group the third and fourth bar are higher than the second, third and fourth bar in the control group although they represent a lower value (0.86/0.81 exp. group vs. 1.45/1.41/1.43 ctrl group)
- The graphs in Figure 3A and B are not correctly aligned
Authors’ Response: All suggested changes are incorporated in them Figure 3 as recommended by reviewer
2) Conclusion:
To be more concise, I would recommend defining “experimental group” Line 517-18) in more detail, e.g. “patients receiving VR+MI training in addition to routine PT”.
Authors’ Response: The suggested changes are incorporated in the manuscript
3) Language:
3.1 Line 251: replace “familiar” by “familiarize”
Authors’ Response: The suggested changes are incorporated in the manuscript
3.2 Line 328-335: remove “the” before “significant improvement”
Authors’ Response: The suggested changes are incorporated in the manuscript
3.3 Line 478: remove “as”
Authors’ Response: The suggested changes are incorporated in the manuscript
4) Citation 27 is invalid.
Authors’ Response: the citation is updated as suggested